# Effect of Genetic Factors, Age and Sex on Levels of Circulating Extracellular Vesicles and Platelets

**DOI:** 10.3390/ijms24087183

**Published:** 2023-04-13

**Authors:** Valeria Orrù, Francesca Virdis, Michele Marongiu, Valentina Serra, David Schlessinger, Marcella Devoto, Francesco Cucca, Edoardo Fiorillo

**Affiliations:** 1Institute for Genetic and Biomedical Research, National Research Council (CNR), 08045 Lanusei, Italy; 2Laboratory of Genetics and Genomics, National Institute on Ageing, National Institutes of Health, Baltimore, MD 21224, USA; 3Institute for Genetic and Biomedical Research, National Research Council (CNR), 09042 Monserrato, Italy; 4Department of Translational and Precision Medicine, Sapienza University, 00185 Rome, Italy; 5Department of Biomedical Sciences, University of Sassari, 07100 Sassari, Italy

**Keywords:** extracellular vesicles, flow cytometry, GWAS

## Abstract

Extracellular vesicles (EVs) mediate cell interactions in biological processes, such as receptor activation or molecule transfer. Estimates of variation by age and sex have been limited by small sample size, and no report has assessed the contribution of genetic factors to levels of EVs. Here, we evaluated blood levels of 25 EV and 3 platelet traits in 974 individuals (933 genotyped) and reported the first genome-wide association study (GWAS) on levels of these traits. EV levels all decreased with age, whereas the trend for their surface markers was more heterogeneous. Platelets and CD31dim platelet EVs significantly increased in females compared to males, although CD31 expression on both platelets and platelet EVs decreased in females. Levels of the other EV subsets were similar between sexes. GWAS revealed three statistically significant genetic signals associated with EV levels in the *F10* and *GBP1* genes and in the intergenic region between *LRIG1* and *KBTBD8*. These add to a signal in the 3′UTR of *RHOF* associated with CD31 expression on platelets that was previously found to be associated with other platelet traits. These findings suggest that EV formation is not a simple, constant adjunct of metabolism but is under both age-related and genetic control that can be independent of the regulation of the levels of the cells from which the EVs derive.

## 1. Introduction

Extracellular vesicles are small lipid-encapsulated bits of cells secreted into the extracellular space during biological processes such as cellular activation, maturation, proliferation, aging and apoptosis [1]. Recently, they have been shown to participate in intercellular signaling and communication and in modulating both homeostatic and pathological processes [2,3]. Indeed, by transferring nucleic acids and a range of proteins to recipient cells, EVs are implicated in tumor cell invasion and metastasis, metabolic and cardiovascular disease, angiogenesis, immune response, and antigen presentation. The identification and characterization of EVs from multiple body fluids—urine [4], breast milk [5], cerebrospinal fluid [6], blood [7], saliva [8], and bile [9]—thus become important to clarify their function in health and disease.

EVs are heterogeneous in size, cellular origin, and composition and content of proteins, nucleic acids, lipids, surface receptors and glycans. In the blood, EVs can be classified into three major groups: exosomes, microvesicles, and apoptotic bodies. Exosomes ranging in size from 30 nm to 100 nm in diameter are considered intraluminal vesicles generated by the inward budding of the multivesicular body (MVB) membrane and are secreted following the fusion of the MVB with the cell membrane [10]. Microvesicles from 50 nm to 1000 nm in size are formed directly by the outward budding and fission of the plasma membrane. In contrast, apoptotic bodies from 500 nm to 5 μm in diameter are vesicles shed by cells undergoing programmed death [11].

EVs are characterized by markers derived from the cell of origin, but they also contain miRNA, lipid and protein profiles that differ from those of their cell of origin [12]. Interestingly, their content can reflect both physiologic and pathogenic conditions, and indeed, many studies have identified variations in EV counts and content in different human diseases. For instance, changes in EV concentrations have been observed in cardiovascular disease [13]. Elevated levels of platelet-derived microvesicles have also been observed in diabetes mellitus [14]. Other evidence shows that EVs appear at significantly higher levels in systemic lupus erythematosus and rheumatoid arthritis cases than in healthy controls [15]. Moreover, EVs contain different RNA species that have been examined as potential biomarkers for various diseases, including different types of cancer [16,17]. Such biomarkers could be especially useful for pathologies for which invasive diagnostic approaches are currently used.

Flow cytometry, enabling multi-parameter single-particle analysis, is a promising technique to measure EVs. A recent article by Marchisio and colleagues [18] described the EV detection method used in the present study based on lipophilic cationic dye positivity and phalloidin negativity. To demonstrate the correct morphology and size of EVs separated by flow cytometry, they used transmission electron microscopy and control beads with a size similar to EVs. Other articles related to SARS-CoV-2 status have been published using the same EV detection approach [19,20], and several other studies have identified EVs by flow cytometry instead of using classical EV detection methods [21]. Previous studies have measured EVs in samples of a very small size [22], and no study, to our knowledge, has investigated the possible genetic regulation of their levels or has systematically assessed their trends during lifespan or by sex in a large general population cohort.

Here, we measured 28 circulating traits including 25 EV and 3 platelet traits, in up to 974 individuals from a general population and performed a genome-wide association study (GWAS). We also assessed age- and sex-related differences in these traits.

## 2. Results

### 2.1. Age Effects on EV and Platelet Traits

We assessed 28 peripheral blood traits on up to 974 Sardinian volunteers. Twelve traits were absolute counts (AC, including 11 EV subsets and platelets, expressed as events/μL) and 16 were median fluorescence intensities of surface antigens (MFI, 14 assessed on EVs and 2 on platelets), as further described in Section 4 and Appendix A and Figure 1.

All of the absolute counts measured showed a downward trend across the lifespan (Appendix A), which was highly significant for platelet and CD31dim platelet EVs (Figure 2a,b), significant for neuronal and leukocyte EVs (Figure 2c,d), and nominal for endothelial and glial EVs (Figure 2e,f). This is in line with previous observations of EV counts [23,24,25]. We also evaluated the phenotypic correlations between the most represented trait, platelet count, and the levels of the EV subpopulations. We observed a significant positive correlation of platelet levels with platelet EVs; EVs from other cell types showed no significant correlation with platelet counts (Appendix A). 

The reduction of EV levels was not necessarily accompanied by a reduction in the expression of their characteristic surface antigens; in fact, both CD31 and CD41a expression on platelet EVs increased with aging (Figure 2g,h). Importantly, CD31 expression on platelets decreased, while CD41a expression on platelets was quite stable (Figure 2i,j), indicating that EVs have their own idiosyncratic regulation independent of the cell from which they derive. In contrast, glial EVs, which did not significantly decrease with aging (Figure 2f), were characterized by a strong reduction in CD11b expression (Figure 2k).

### 2.2. Sex Effects on EV and Platelet Traits

Sex significantly affected platelet traits: both platelets and CD31dim platelet EVs increased in females compared to males (Figure 2a,b), although levels of the other EV subsets were similar between sexes. Regarding the expression level of the surface antigens, CD31 was reduced in females compared to males in both platelets (Figure 2i) and platelet EVs (Figure 2g,l) and, to a lesser extent, in endothelial EVs (Appendix A). The other proteins measured on platelet and EV surfaces showed no significant differences between males and females.

### 2.3. Genome Wide Association Study of EV and Platelet Traits

We performed a GWAS that, correcting for the number of the assessed traits, had a *p*-value for statistical significance of 2.47 × 10^−10^ (see Section 4, Table 1). We identified one significant signal led by SNP rs11553699 in the 3′UTR of the *RHOF* and *TMEM120B* genes on chromosome 12, associated with the expression of CD31 on platelets (*p*-value = 1.97 × 10^−15^, beta = 0.46) (Figure 3a). This variant was also an s-QTL (splicing quantitative trait locus) for *TMEM120B* [26]; however, the role of this gene in platelet regulation or function is not known. RHOF belongs to the GTPase family, which is critical for platelet function. It is expressed in platelets and seems to interact with cytoskeleton regulators, but no role in platelets has been proposed [27].

Three novel signals did not reach the *p*-value of 2.47 × 10^−10^ but met the classical genome-wide significance threshold of 5.00 × 10^−8^ and were associated with the level of EV subsets. The first signal, led by rs3213004 in the intronic region of the *F10* gene on chromosome 13, was associated with neuronal EVs (*p* = 2.47 × 10^−8^, beta = 0.51) (Figure 3b). F10 is the vitamin K-dependent coagulation factor X of the blood coagulation cascade, which, when activated, converts prothrombin to thrombin.

The second signal, led by rs116060256 in the intergenic region between the *LRIG1* and *KBTBD8* genes on chromosome 3, was associated with CD171hi EVs (*p* = 2.51 × 10^−8^, beta = −0.50) (Figure 3c). These two genes are involved in neural precursor cell proliferation (*LRIG1*) [28] and neural crest specification (*KBTBD8*) [29].

The third signal was led by a missense variant, rs148526074 (c.C793A:p.Q265K), in exon 6 of the *GBP1* gene (chromosome 1) and was associated with leukocyte EV counts (*p* = 2.80 × 10^−8^, beta = 1.02) (Figure 3d). This gene belongs to the group of guanylate-binding proteins and is involved in inflammasome activation following microbial insults [30,31].

## 3. Discussion

Previous studies used a very small sample size [22] that precluded the possibility of performing GWAS and often focused on a specific EV subtype or on EVs in toto [23]. The present study assessed the genetic regulation of EVs for the first time.

Initially, EVs were isolated by ultracentrifugation and density gradient flotation or other time-consuming approaches [32,33,34]. More recently, they began to be detected by alternative approaches, such as flow cytometry, staining biological liquids (blood, milk, tears, saliva) with two reagents: a lipophilic cationic dye that diffuses into EVs by their membrane potential, and phalloidin, which interacts with cytoskeleton actin only when EV membranes are damaged, and thus does not bind intact EVs (see Section 4). This procedure enriches EVs for further subsequent subset detection using surface markers that characterize the cells/fragments from which EVs derive [18,19,20]. This approach identifies EVs more rapidly and easily than previously used, facilitating the processing of the large number of samples necessary to perform GWAS. However, because profiling and enumeration of EVs usually requires two or more complementary techniques to aid in their definitive identification, controls to verify their size are always recommended. Furthermore, EV detection by flow cytometry also needs to be improved because the EV area is often enriched with non-EV fragments that can hinder correct EV counting.

Here, we assessed about 1000 individuals, still relatively small for GWAS, but sufficient for an exploratory analysis that already yielded suggestive results. Thus, a larger number of samples should provide additional and more robust genetic signals.

Among the 4 signals we described, the most significant was in the 3′UTR region of *RHOF,* associated with the expression of CD31 in platelets. This variant was already known to be strongly associated with platelet volume, count and distribution width [35]; thus, the identified association can be considered a proof of concept of the method using flow cytometry to measure small molecules, such as EVs and platelets.

The other three novel genetic signals included an association of neuronal EVs with the *F10* gene, consistent with the suggested involvement of coagulation factors, such as F10, in neurodegeneration [36,37,38]. A second signal linked CD171hi EVs, which are mainly derived from the nervous system, with *LRIG1* and *KBTBD8*, both genes involved in neuronal development and differentiation [39]. The last signal associated leukocyte EVs, known to induce immune and inflammatory responses [40], with *GBP1*, a gene involved in the inflammatory response. All of these genes are correspondingly implicated in the function of the EV subsets with which they are associated.

Given the global increase in lifespan, there is growing interest in identifying circulating factors that correlate with biological or chronological age. Because EVs appear to undergo alterations in number and internal content during senescence [41], we tested the correlation of EV subsets with age. We observed a general reduction in the levels of EV subsets, suggesting that the EV-mediated interaction of cells may decline with age. This might cause an increased risk for several diseases more common in the elderly. Indeed, neural stem cell-derived EVs have been suggested to have a neuroprotective function by reducing reactive oxygen species and inflammatory cytokines [42] and may thus have therapeutic potential in neurodegenerative disorders. Furthermore, native and engineered EVs have been suggested for the treatment of different brain pathologies [43].

On the other hand, it has been observed that neuronal EVs derived from Parkinson’s patients and transferred to mice may exacerbate the disease in a Parkinson mouse model [44]. Furthermore, in the area of blood banking, extracellular vesicles derived from red blood cells and accumulated during blood storage might have proinflammatory and procoagulant effects and be implicated in adverse transfusion events and posttransfusion outcomes [21,45]. Therefore, while the reduction of circulating EVs may result in detrimental reduction of communication between cells, an increase in EV levels could still be harmful depending on their cargo. Any therapeutic interventions based on EVs thus remain conjectural, requiring further study.

In contrast to the general downtrend of EV levels with age, the expression tendencies of their surface proteins during aging were less homogeneous, with some of them increasing and others decreasing. For example, the levels of glial EVs, characterized by CD11b positivity, did not significantly decrease during aging, but the expression of CD11b on their surface was strongly reduced. This may indicate that aging primarily causes reduced levels of circulating EVs, but when such a reduction does not occur, energy is saved by a decreased production of surface proteins.

Furthermore, sex-specific dissection of EVs (and, more generally, of any quantitative trait involved in disease susceptibility or providing a therapeutic target) is relevant to better understand diseases with a disproportionate incidence in males and females. It can also help to assess correct dosages of drugs. Considering the growing interest in the use of EVs as therapeutic targets [46], we analyzed sex-specific differences in EV levels but observed a significant impact of sex only on those derived from platelets.

Overall, we infer that EVs are not simply mirrors of the cells/fragments from which they are derived. This is clearly demonstrated by the increase of CD31 on platelet EVs during the lifespan, while the same marker decreased in platelets. We also infer that markers are specifically regulated on different EV subtypes, e.g., over the lifespan, CD31 changes in platelet EVs but not in endothelial EVs.

Thus, EVs can be treated like any other component measured in blood, such as neurofilaments or fatty acids, whose increasingly relevant roles in disease susceptibility may provide therapeutic targets or prognostic/diagnostic markers [47,48,49]. EV assessment in large cohorts could further analyze their involvement in complex diseases through the colocalization of GWAS signals for EV levels and diseases and Mendelian randomization [50].

## 4. Materials and Methods

### 4.1. Samples

This study was approved by the Sardinian Regional Ethics Committee (protocol no. 2171/CE) and all participants provided written informed consent. The 974 participants (aged 23–92 years, 61% females) belonged to the general population SardiNIA cohort originating from four towns located on the central east coast of Sardinia, Italy [51]. Of 974 volunteers, 933 have been genetically characterized for about 18.4 million variants [52].

### 4.2. Sample Preparation and Sample Staining

Fresh peripheral blood from volunteers was collected into sodium citrate tubes. The tubes were inverted five times and processed within 2 h of blood drawing to minimize time-dependent artifacts. To obtain absolute cell counts, TruCount absolute counting tubes were used (BD Biosciences, San Jose, CA, USA, Cat#340334). For staining whole peripheral blood, 95 μL of filtered phosphate buffered saline (PBS, Sigma, St. Louis, MO, USA, Cat#D8537; 0.22 μm filter unit; Millipore, Carrigtwohill, Ireland, Cat#SLAGP033RB) was dispensed in BD Trucount tubes without disturbing the bead pellet. Five microliters of blood was added using reverse pipetting. Characterization of extracellular vesicles was carried out using two custom kits named “Dye Integer EV Detection Kit” (BD Biosciences, Cat#626267) and “CD41a/CD31/CD45 EV Detection Kit” (BD Biosciences, Cat#626266) that are not present in the standard catalogue but are available on demand. The first kit is based on lipophilic cationic dye that diffuses into EVs by their membrane potential; the kit also contains phalloidin FITC, which binds the cytoskeleton protein actin only in particles having damaged membranes. We then prepared a reagent mix, as described in Appendix A. To eliminate crystal molecules, the mix was spun down at 16,000× *g* for 20 min immediately before use. We then added 100 μL of the mix to the TruCount tubes containing 100 μL of diluted sample (95 μL of PBS and 5 μL of blood). After incubation for 45 min in the dark at room temperature, 1.5 mL of filtered PBS was added to the samples and finally acquired by a FACSCanto II (BD Biosciences) flow cytometer and analyzed by FACSDiva software, version 8.0.1 (BD Biosciences). To eliminate the background, we used a threshold on the fluorescence channel of the lipophilic probe. However, because events below 100 nm (such as exosomes) could not be distinguished from the background, with the threshold used, we eliminated both the background and the exosomes; thus, exosomes were not considered in this work.

### 4.3. Gating Strategy Analysis

Phalloidin negative and lipophilic cationic dye positive events were considered enriched in EVs, as previously described [18]. Leukocyte-derived EVs were identified as CD45 positive and further subdivided into CD45dim EVs and CD45high EVs. CD31 and CD41a were evaluated on CD45-negative EVs to define EVs derived from endothelial cells (CD31+ CD41a−) and platelets (CD31+ CD41a+). Within CD45 negative EVs, we also used the CD11b marker to identify glial-derived EVs and the neuronal surface marker CD171 to characterize EVs of neuronal origin. Outside the EV area, platelets were identified based on their positivity for both CD31 and CD41a markers. Red blood cells were identified by their morphology and lipophilic cationic dye positivity and were not acquired to avoid storing heavy files.

To evaluate the correct size (50 nm 1000 nm) of the events acquired in the EV area, we performed set-up experiments using a mix of beads with the following diameters: 100, 300, 500, and 900 nm, called Megamix-Plus FSC (Biocytex, Marseille, France, #7802), which have been specifically generated for cytometer settings in microparticle analysis (Figure 1i–j). A further internal control was the parallel acquisition of platelets, which measure 1500–3000 nm in diameter; indeed, as we observed in Figure 1d, the EVs were always smaller than platelets.

### 4.4. Statistical Analysis

#### 4.4.1. Genotyping and Imputation

Genetic analyses were performed on 933 samples genotyped with OmniExpress Illumina array, as previously described for the entire SardiNIA cohort [52]. Imputation was performed on a genome-wide scale using a Sardinian sequence-based reference panel of 3514 individuals and Minimac software, version 4.0 [53] on pre-phased genotypes [52]. After imputation, only markers with imputation quality (RSQR) >0.3 for estimated MAF ≥ 1% or >0.6 for MAF < 1% were retained for association analyses [54], yielding 18,425,755 variants (17,209,121 SNPs and 1,216,634 indels) useful for analyses.

#### 4.4.2. Association Analysis

Genome-wide association analysis for each quantitative trait was carried out using the q.emmax (quantitative EMMAX–Efficient Mixed Model Association eXpedited) function included in EPACTS-3.2.6 (https://genome.sph.umich.edu/wiki/EPACTS, accessed on 10 July 2014). The method implemented in this software accounts for a wide range of sample structures, such as cryptic relatedness and population stratification, by applying a linear mixed model adjusted for a genomic-based kinship matrix obtained from quality-checked genotyped autosomal SNPs with MAF > 1% [52]. All assessed traits were normalized with inverse-normal transformation and adjusted for sex and age as covariates. To adjust for multiple testing, Bonferroni correction was applied to the empirical significance threshold (*p* = 6.91 × 10^−9^) by considering the total number of absolute cell counts and MFIs assessed here (*n* = 28), yielding a final threshold of *p* < 2.47 × 10^−10^. Variants in the sex chromosomes were not analyzed.

#### 4.4.3. Age, Sex, and Correlation Analysis

To assess the impact of age and sex on the 28 traits measured, we applied a linear regression model and used R software, version 4.2.2. Traits were normalized before the analyses using inverse normal transformation. The significant threshold of 1.79 × 10^−3^ was obtained by dividing the nominal *p*-value of 0.05 for the number of traits assessed (*n* = 28). Phenotypic correlations between platelet and EV levels were calculated using Spearman’s coefficient in R software, version 4.2.2. The significant threshold of 4.55 × 10^−3^ was obtained by dividing the nominal *p*-value of 0.05 for the number of traits expressed assessed (*n* = 11 EV absolute counts vs. platelet count).

## Figures and Tables

**Figure 1 ijms-24-07183-f001:**
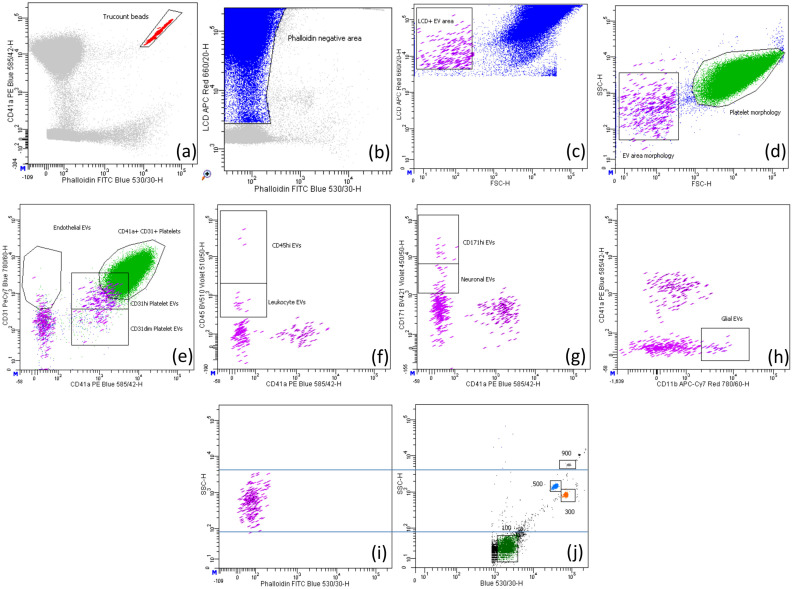
Gating strategy to identify EVs and platelets. (**a**) Counting beads (red) were identified based on their high fluorescence for FITC and PE fluorochromes; (**b**) the phalloidin negative area (blue) contained both extracellular vesicles and platelets. The EV area (violet) was identified by intersecting the lipophilic cationic dye (LCD) positive EV area in (**c**) with the EV area morphologically assessed in (**d**). Platelets (green) were identified by intersecting platelets morphologically detected in (**d**) with those identified by CD31 and CD41a positivity in (**e**). (**e**) EVs double positive for CD31 and CD41a were considered of platelet origin and further divided into CD31dim and CD31hi; EVs positive for CD31 but negative for CD41a were considered of endothelial origin. EVs positive for CD45, CD171 and CD11b were considered leukocyte (**f**), neuronal (**g**), and glial (**h**) origins, respectively. In (**f**,**g**), we also gated CD45hi and CD171hi events, respectively, that could be either EVs expressing high levels of these specific markers or crystals. The comparison between the size of EVs (**i**) and of the Megamix beads (**j**) is shown.

**Figure 2 ijms-24-07183-f002:**
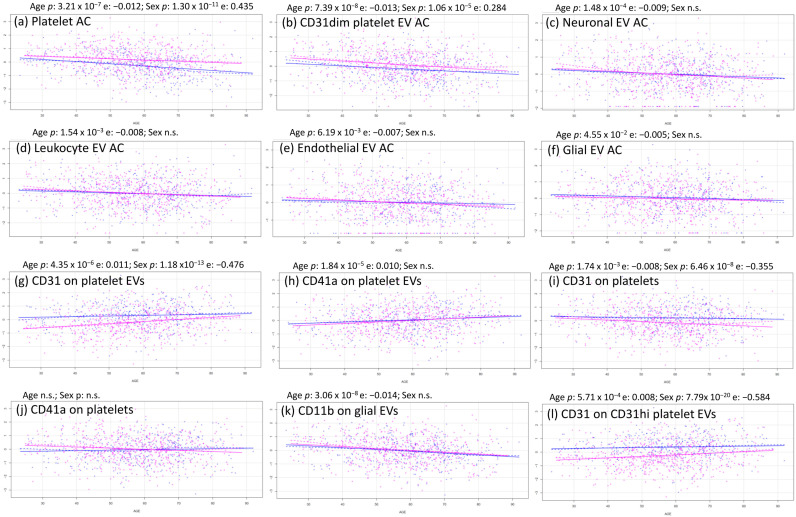
Most relevant age–sex tendencies of EVs and platelets. Each scatter plot represented males in blue and females in pink, age in the *x* axis, and normalized trait in the *y* axis.

**Figure 3 ijms-24-07183-f003:**
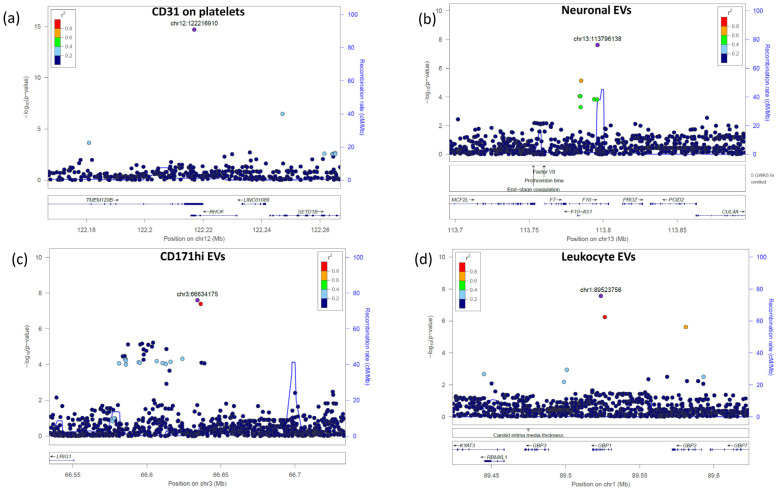
Regional association plots of the most significant identified signals. The significance of the association (−log10[*p* value]; left *y* axis) for each trait was plotted relative to the genomic positions on the hg19/GRCh37 genomic build (*x* axis). SNPs were colored to reflect their linkage disequilibrium with the top SNP (indicated with a purple dot and with its genomic position at the built GRCh37/hg19).

**Table 1 ijms-24-07183-t001:** GWAS results for EVs and platelets. *p*-values lower than 5.00 × 10^−8^ were shown. Beta + referred to the alternative (Alt) allele. Minor allele frequency (MAF); Standard error of beta (SE beta); Reference allele (Ref).

Trait_Name	rsID	Position (GRC37-hg19)	MAF	*p*-Value	Beta	SE Beta	Ref	Alt	Major	Minor	Gene Function	Gene
CD31 on platelets	rs11553699	12:122216910	0.21257	1.97 × 10^−15^	0.4563	0.05644	A	G	A	G	UTR3	*RHOF,TMEM120B*
Neuronal EV AC	rs3213004	13:113796138	0.06424	2.47 × 10^−8^	0.5147	0.09152	G	A	G	A	intronic	*F10*
CD171hi EV AC	rs116060256	3:66634175	0.07232	2.51 × 10^−8^	−0.5035	0.08958	G	A	G	A	intergenic	*LRIG1,KBTBD8*
Leukocyte EV AC	rs148526074	1:89523756	0.01634	2.80 × 10^−8^	1.021	0.1824	G	T	G	T	exonic	*GBP1*
CD171+ EV AC	rs3213004	13:113796138	0.06424	2.92 × 10^−8^	0.5119	0.09152	G	A	G	A	intronic	*F10*
CD171hi EV AC	rs17777828	3:66636360	0.07126	3.97 × 10^−8^	−0.4974	0.0898	A	G	A	G	intergenic	*LRIG1,KBTBD8*

## Data Availability

Not applicable.

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
