# Peer review of "Effect of Genetic Factors, Age and Sex on Levels of Circulating Extracellular Vesicles and Platelets"

_ijms, 2023, doi:10.3390/ijms24087183_

Round 1
Reviewer 1 Report
The manuscript of Valeria Orrù et al. described that the effects of aging, sex and genomic information for the amount of EVs in blood. They analyzed blood samples from 974 individuals using flowcytometry. Although it has been thoroughly experimented, it is insufficient to determine by only one experimental method whether the collected fraction is truly EVs. The authors should confirm it. Therefore, I do not recommend this manuscript for publication in International Journal of Molecular Science in its current form.
Major concerns
1. The authors should confirm EVs using other conventional methods ex) TEM, AFM, NTA, corresponding to the previous report (Théry et al., J. Extracell. Vesicles, 2018).
2. In material and methods, the authors described about collection and detection of EV using custom kits. Is the “Dye Integer EV Detection Kit” available to general users? I could not find the catalog number in the catalog of BD Bioscience.
3. In figure 1j-k, the size of phalloidin negative population is over 100 mm. Did the authors exclude small extracellular vesicles (exosomes) in phalloidin negative population? If so, the authors should explain the reason in manuscript.
4. In Figure 2, it seems that the amounts of EVs derived from some cell types are inversely correlated with age. However, the analysis seems insufficient. The authors should check whether platelet counts and non-platelet-derived EVs are correlated in individual blood data. Is there a correlation between the number of platelets and the number of each CD41a (platelet EV) /CD45 (leukocyte EV)/CD171 (neuronal EV)/CD11b (glial EV)-positive EVs?
Minor concerns
1. Please add abbreviation of absolute count (AC) in figure legend or text.
Reviewer 2 Report
In the manuscript entitled "Effect of Genetic Factors, Age and Sex on Levels of Circulating Extracellular Vesicles and Platelets" Valeria Orrù et al. provide evidence regarding the association of genetic factors and EVs and platelets using GWAS analysis and a significant amount of test subjects. The manuscript reads well, and the study is focused.
This Reviewer has some comments, as follow:
1. The authors use a detailed protocol of flow cytometry (with extensive gating strategy) to assess EVs parameters. The protocol seems fine but since EV profiling and enumeration usually needs 2 or more complimentary techniques due to the difficulty of the procedure (and especially for EV smaller than 250-300 nm) this reviewer believes that a small piece information regarding this issue must be added to the manuscript (perhaps as a limitation or a comment).
2. The authors’ findings are very informative but this reviewer believes that the manuscript could be strengthened if a more detailed discussion regarding the authors’ findings (especially in the case of age differences) is added. At this point it would also be very interesting if a small piece discussion was added focusing on EV generation differences between in vivo and accelerated aging (namely blood bank conditions).
3. Figure legends 1: This author believes that it does not read so well since the usage of subpanel indicators (e.g., e), d), etc.) is a bit confusing. I would recommend using (a), (b) etc. when referring to a specific subpanel and a), b) when starting its description.
4. In general, a slight proof-reading for minor mistakes throughout the manuscript might be needed.
Round 2
Reviewer 1 Report
The authors have now resubmitted a revised version of the manuscript taking into account concerns raised by the reviewers. The new version of the manuscript is improved and the conclusions are better supported.
Reviewer 2 Report
The authors have addressed my comments. Thus, I have nothing more to add.